# Novel Double Hybrid-Type Bone Cements Based on Calcium Phosphates, Chitosan and Citrus Pectin

**DOI:** 10.3390/ijms241713455

**Published:** 2023-08-30

**Authors:** Piotr Pańtak, Joanna P. Czechowska, Ewelina Cichoń, Aneta Zima

**Affiliations:** 1Faculty of Materials Science and Ceramics, AGH University of Science and Technology, Mickiewicza Av. 30, 30-058 Krakow, Poland; 2Jerzy Haber Institute of Catalysis and Surface Chemistry, Polish Academy of Sciences, Niezapominajek 8, 30-239 Krakow, Poland; ewelina.cichon@ikifp.edu.pl

**Keywords:** bone cements, hybrid materials, calcium phosphate, chitosan, dual setting, injectability, polysaccharides

## Abstract

In this work, the influence of the liquid phase composition on the physicochemical properties of double hybrid-type bone substitutes was investigated. The solid phase of obtained biomicroconcretes was composed of highly reactive α-tricalcium phosphate powder (α-TCP) and hybrid hydroxyapatite/chitosan granules (HA/CTS). Various combinations of disodium phosphate (Na_2_HPO_4_) solution and citrus pectin gel were used as liquid phases. The novelty of this study is the development of double-hybrid materials with a dual setting system. The double hybrid phenomenon is due to the interactions between polycationic polymer (chitosan in hybrid granules) and polyanionic polymer (citrus pectin). The chemical and phase composition (FTIR, XRD), setting times (Gillmore needles), injectability, mechanical strength, microstructure (SEM) and chemical stability in vitro were studied. The setting times of obtained materials ranged from 4.5 to 30.5 min for initial and from 7.5 to 55.5 min for final setting times. The compressive strength varied from 5.75 to 13.24 MPa. By incorporating citrus pectin into the liquid phase of the materials, not only did it enhance their physicochemical properties, but it also resulted in the development of fully injectable materials featuring a dual setting system. It has been shown that the properties of materials can be controlled by using the appropriate ratio of citrus pectin in the liquid phase.

## 1. Introduction

Calcium phosphate cements (CPCs) are self-setting ceramic materials commonly used in orthopedic applications. The setting process of α-tricalcium phosphate (α-TCP) containing CPCs is based on its hydrolysis to non-stoichiometric hydroxyapatite in ambient or elevated temperature [1]. CPCs are well known to be biocompatible due to their chemical similarity to the inorganic part of bone [2,3]. Lately, a new group of chemically bonded materials, namely biomicroconcretes, composed of aggregates, e.g., in the form of granules, microspheres or pellets that are embedded in the CPC matrix, has been studied. For example, Russo et al. [4] showed the beneficial influence of carbon nanostructures on the mechanical performance of polymeric composites. The aggregates, similar to those in classic concrete, are supposed to cause microcrack retention. Additionally, granules can enrich material with agents exhibiting antibacterial or bactericidal properties, such as polymers (i.e., chitosan, methylcellulose), ions (i.e., Ag^+^, Cu^2+^) or nanoparticles (such as AgNPs and AuNPs) [5,6,7]. It is also possible to include active agents in hybrid materials to overcome issues related to bacterial resistance to antibiotics, as showed previously in the study of De Santis et al. [8].

Despite the excellent bioactivity of calcium phosphate-based chemically bonded biomaterials, their injectability is often poor, which hinders their application in minimally invasive surgery. Moreover, their mechanical properties are not sufficient for load-bearing applications. The improvement in surgical handiness of bone cements can be achieved by introducing polymeric additives, which plasticize the paste and facilitate its application [9,10]. The use of polymers in combination with inorganic calcium phosphates can additionally lead to the formation of organics-in-inorganics hybrids [11]. According to the International Union of Pure and Applied Chemistry (IUPAC) definition, hybrid material is *material composed of an intimate mixture of inorganic components, organic components, or both types of components* [12]. It is believed that in hybrids, the combination of components occurs at the molecular level. For this reason, two classes of hybrids are distinguished. Class I components act together with weak bonds (van der Waals, electrostatic, and hydrogen bonds), while class II components are bonded by strong (covalent or ionic-covalent) chemical bonds [12]. The formation of hybrids in a material’s structure can change the properties of the final product, for example by enhancing the mechanical properties of bone cements [13]. A new trend in reinforcing CPCs takes advantage of the interactions between oppositely charged polyelectrolytes such as chitosan and pectin [14]. Chitosan is a linear polycationic polysaccharide obtained by chemical processing of shellfish industry waste [15]. It possesses excellent bioactivity as well as antimicrobial proprieties, so it is commonly used as a compound for wound dressings, membranes or drug carrier systems [16], while citrus pectin is an anionic polysaccharide derived from citrus pectin peel from food industry waste [17]. From a medical point of view, pectin has important features, including its gelling capacity, biocompatibility, and anti-inflammatory and anticancer properties [18]. In the biomaterials industry, pectin can be used in different forms—coatings, films, hydrogels, conjugates, sprays and many more; however, its application in calcium phosphate cements is still under evaluation [19].

The electrostatic interactions occurring between polycation (chitosan) and polyanion (pectin) lead to the formation of class I hybrids. Due to the different nature of the charges, the selection of these natural polymers results in the formation of a rigid gel [20,21]. This phenomenon has been previously applied to obtain membranes, tissue scaffolds and drug delivery systems. De Almeida et al. [22] studied pectin/gold nanoparticles/chitosan superabsorbent hydrogels with a satisfactory elastic modulus for drug delivery purposes. Martins et al. [23] developed water-stable and mechanically resistant membranes based on pectin and chitosan polyelectrolyte complexes, without covalent crosslinking. However, to the best of our knowledge, the simultaneous application of chitosan and pectin in CPCs has not yet been examined [24]. Due to the structure and properties of natural polymers such as chitosan or pectin, they are susceptible to chemical and physical modifications. Such modifications are carried out to change the properties of the initial polymers by, for example, developing their surfaces or functionalizing the functional groups of their individual monomers [25,26,27].

The aim of this study was to develop, obtain and study new hybrid injectable biomicroconcretes and to determine the effect of citrus pectin introduced in the liquid phase on the physicochemical, applicational and biological properties of those materials. Biomicroconcretes composed of highly reactive α-TCP powder as setting phase, hybrid hydroxyapatite/chitosan (HA/CTS) granules as aggregates, and different liquid phases containing citrus pectin were obtained and investigated. This choice of material composition would allow developing double-hybrid bone substitutes (hybrid in granules, as well as hybrid originated from polyelectrolyte interactions) with a dual-setting mechanism. According to our knowledge, these are the first studies regarding the double-hybrid system in cementitious materials. Other authors have used different dual-setting mechanisms in bone cements. Geffers et al. [28] developed dual-setting brushite-silica gel cements characterized by improved mechanical properties, while Christel et al. [29] studied calcium phosphate-polymethacrylate cements, which were characterized by significantly shortened setting times and improved mechanical strength. However, through careful selection of the liquid phase, it will be possible to obtain fully injectable materials containing hybrid granules for the first time. What is more, due to the use of natural polymers as a modifier of CPCs, this research will pave the way to further modification of similar materials through their functionalization. In addition, appropriate selection of the liquid phase will allow one to obtain materials characterized by setting times characteristic of calcium phosphate-based bone cements.

## 2. Results and Discussion

### 2.1. Setting Times

In clinical practice, cementitious bone substitutes should be characterized by setting times that allow their application by the surgeon to the targeted defect site. This parameter determines how long it takes for the material to set after the liquid and powder phases have been mixed. Optimally, the application should take place during “dough” time, i.e., between the initial and final setting times [30]. Ideally, the initial setting time (t_i_) should be approximately 15 min, whereas the final setting time (t_f_) should be under 30 min [31,32]. However, in the literature, ranges between 4–8 min for t_i_, and up to 15 min for t_f_ also can be found [33].

It is very important that, in the case of materials with complex compositions, the setting time is not adversely affected by the additives. Thus, the choice of components is crucial when designing cementitious biomaterials, especially injectable ones. In our studies, despite the complex compositions of the biomicroconcretes, their setting times were within an acceptable range (Table 1). However, both the initial and final setting times of the cement pastes were strongly dependent on their liquid phase composition.

Setting times of MC1, where the liquid phase constituted solely of 2.0 wt% Na_2_HPO_4_ solution, were the shortest and equal to 4.5 min (t_i_) and 7.5 min (t_f_). In this case, the setting process of biomicroconcrete was based only on the hydrolysis of α-tricalcium phosphate to calcium-deficient hydroxyapatite (CDHA), according to the following chemical equation (Equation (1)) [34,35].
(1)$$3{Ca}_{3}{\left(PO_{4}\right)}_{2}+H_{2}O\to {Ca}_{9}(PO_{4})5(HPO{)}_{4}OH$$

It is known that the rate of α-TCP hydrolysis to CDHA depends on many external and internal factors, including: physical conditions during paste formation (e.g., temperature, humidity), the presence of a setting accelerator (e.g., Na_2_HPO_4_) or inhibitor, as well as the kinds and amounts of polymeric additives. Usually, the addition of polymer to CPCs causes an increase in setting times but, depending on the polymer type, various processes can be responsible for this phenomenon [36]. In the case of materials containing citrus pectin (MC2–MC5), α-TCP hydrolysis is also a second setting reaction, which is directly connected with pectin cross-linking by Ca^2+^ ions that occur [14]. The double-setting mechanism resulted in longer setting times, because the process of α-TCP hydrolysis was disrupted by calcium-mediated cross-linking of the citrus pectin. It was observed that the materials’ setting times increased proportionally to the amount of citrus pectin gel in the liquid phase and were in the range of 9.0–17.5 min (t_i_) and 16.5–32.0 min (t_f_). Similar results were observed elsewhere [37] for low esterified pectin from citrus peels and apple pomace added to the solid phase of cementitious materials. The setting times of biomicroconcretes in which the liquid phase consisted solely of citrus pectin gel (MC5) were the longest—30.5 min for the initial and 55.5 min for the final setting time, respectively. This was probably connected with the water uptake by citrus pectin, which inhibited the hydrolysis process of tricalcium phosphate, whereas the amount of crosslinked pectin was too small to allow the material to set and maintain its integrity during the measurements. As a result, the material MC5 was excluded from further studies due to exceeding the setting times recommended by clinical practice. The polymeric additives in calcium phosphate-based bone cements acted also as plasticizers. By altering the rheological properties of the paste, pectin favors its injectability.

### 2.2. Injectability

The injectability, in addition to the setting times, is a key property for cementitious-type bone substitutes, especially for non-invasive surgical applications [38]. Currently, a standard for the injectability of calcium phosphate bone cements is not available. However, based on the literature, the maximum force on the syringe plunger during paste extrusion is determined as 100 N [39]. This force determines the value that the surgeon applies to the syringe plunger with one hand during the surgical procedure. The results of injectability tests and the appearance of the cement pastes after injection directly into SBF are shown in Figure 1.

The use of citrus pectin allowed for sufficient plasticization of the paste, improving its viscosity. In opposition to CPCs, which are usually non-injectable [40], the obtained biomicroconcretes containing citrus pectin (MC2–MC4) were fully injectable and maintained their cohesion after being extruded into SBF. Phase separation was observed only in the case of MC1, which did not contain added citrus pectin in the liquid phase. The other pastes did not show phase separation during injectability tests. It was also noticed that the force required to inject biomicroconcretes decreased with the increasing amount of citrus pectin. For materials MC3 and MC4, the force was lower than the recommended 100 N [39]. Similar results were obtained by Arkin et al. [41], where a polymer in the form of carboxymethyl cellulose was used as the injectability provider. The presence of citrus pectin in the liquid phase resulted in excellent cohesion of the MC2–MC4 materials, which allowed them to retain their shape in contact with SBF. The double-setting mechanism including simultaneous hydrolysis of α-TCP to CDHA and calcium-mediated crosslinking of citrus pectin also contributed to the paste cohesion [42,43]. The effect on the injectability of calcium phosphate-based cement pastes is an outcome of their rheological properties. Self-setting materials change their rheological properties over time. Bone cements based on calcium phosphates and biopolymers show a pseudo-thixotropic character [44].

### 2.3. Phase Composition and Structural Studies

Calcium phosphate ceramics are used in biomaterial engineering due to their similarity in phase composition to that of inorganic bone. Through phase and structural testing, that relationship can be confirmed. The initial α-TCP powder was composed of α-TCP phase (97.0 ± 1.0 wt%) and a small amount of hydroxyapatite (3.0 ± 1.0 wt%), whereas the HA/CTS hybrid granules were composed of HA as the only crystalline phase. In biomicroconcretes, XRD analysis revealed the presence of two crystalline phases: α-TCP and hydroxyapatite, and their proportion varied according to the setting environment (air/SBF) (Figure 2A). Moreover, the amorphous halos originating from polymers were present.

The detailed phase compositions of the obtained materials are presented in Table 2.

The materials containing citrus pectin in the liquid phase were characterized by slightly slower hydrolysis of α-TCP to hydroxyapatite. The amount of α-TCP phase in the materials after 7 days in air was in the range from 54 ± 1 to 61 ± 1 wt%, while the proportion of the hydroxyapatite phase was in the range from 39 ± 1 to 46 ± 1 wt.

Slower hydrolysis of α-TCP may be explained by water absorption by CP [45,46]. The analysis of the samples’ phase compositions after 7 days of incubation in SBF confirmed that α-TCP, as a thermodynamically metastable phase, spontaneously hydrolyzed to calcium-deficient hydroxyapatite. Similar observations were previously reported for α-TCP based bone cements [21,33].

The materials’ FTIR spectra (Figure 2B) after setting and hardening showed characteristics bands at ~598, ~559 cm^−1^ (bending), ~965, and ~1020 cm^−1^ (stretching), which corresponded to the vibrations of PO_4_^3−^ groups. The wide band in the range of ~3000–3800 cm^−1^ was assigned to the absorbed water. Moreover, the spectra possessed an absorption band at ~870 cm^−1^ that corresponded to HPO_4_^2−^ groups, confirming the presence of non-stoichiometric hydroxyapatite. Additionally, in a similar spectral range (~873–875 cm^−1^), carbonate bonds may have appeared in the material, whereas the existence of a band at 1424 cm^−1^ indicated partial substitution of CO_3_^2−^ in the hydroxyapatite structure. Similar results were obtained for calcium phosphate-based biomaterials by Gao et al. [47] and Cichoń et al. [48]. FTIR studies confirmed the presence of chitosan and pectin in the biomaterials. The absorption band with a maximum at ~2930 cm^−1^ can be attributed to alkyl C-H (stretching) vibrations. The band at around 1649 cm^−1^ was assigned to N-H bending vibrations of the primary amine and confirmed the presence of chitosan and amidated citrus pectin in the materials. What is more, the bands around 1315 cm^−1^ and 3573 cm^−1^ corresponded to C-N and O-H stretching vibrations, respectively. For the pectin- and chitosan-containing systems, the formation of electrostatic and/or hydrogen bonding is possible. In the biomicroconcretes, the creation of polyelectrolyte complexes at the hybrid granule/pectin interface took place. Polyelectrolyte complexes between these molecules were previously studied inter alia by Rashidova et al. [49] as well as Dziadek et al. [14]. The presence of electrostatic interactions between components of developed biomicroconcretes allows the creation of a double hybrid system, thus influencing other physicochemical properties of the biomaterials. FT-IR spectroscopy, as a complementary research method, confirmed the XRD results and demonstrated the presence of both polymers, which could not be easily confirmed using diffractometry alone. Moreover, the similarity of the obtained materials to the inorganic part of human bone was also acknowledged.

### 2.4. Microstructure

By observing the microstructure of the developed materials, we were able to assess their suitability for potential use as a bone tissue substitute. In addition, observations enabled us also to identify the presence of microstructural defects. Scanning microscopy (SEM) observations of the obtained biomaterials were performed both after 7 days of setting and hardening in air (Figure 3), as well as after 7 days of sample incubation in SBF.

The biomicroconcretes possessed a compact and homogeneous microstructure formed by an α-TCP matrix and hybrid hydroxyapatite-chitosan granules. Materials MC2–MC4 where citrus pectin was added were characterized by more intimate contact between hybrid HA/CTS granules and the cementitious matrix, as the formed polymeric bridges linked the mentioned components. The highest number of bridges was observed in the material MC4, with a maximal amount of CP in the liquid phase. Analogous bridges were previously observed by Czechowska et al. [21] where chitosan was introduced in the liquid phase of calcium phosphate bone cement. The presence of polymeric bridges was related not only to the hybrid HA/CTS granules, but also to the electrostatic interactions between the polycationic chitosan contained in the granules and the polyanionic citrus pectin. The confirmation of interaction between the oppositely charged polymers could also be confirmed by the increased presence of polymer bridges in the material. In addition, the presence of such bridges could affect the mechanical strength of the materials. Polymer bridges can potentially reinforce the material in a manner similar to the fibers commonly used in composites.

### 2.5. Compressive Strength

Mechanical strength is an important parameter when assessing materials. From the clinical point of view, the mechanical strength of these substitutes must be within the range of the tissue they would be replacing. In the case of ceramic materials such as CPCs, compressive strength is the most commonly measured parameter to determine their mechanical properties [50]. The results of compressive strength measurements of the obtained materials carried out after 7 days of setting and hardening are shown in Figure 4.

It was found that the compressive strength of MC1 samples, where the liquid phase was based only on 2 wt% Na_2_HPO_4_ solution, was the lowest (5.75 ± 1.08 MPa). The presence of citrus pectin in the liquid phase of biomicroconcretes significantly improves the mechanical properties of the final materials up to 13.24 ± 1.28 MPa (MC4). Furthermore, the increase in mechanical strength with the increasing amount of pectin was noticed. A similar correlation was observed previously elsewhere [37]. The polymer may provide mechanical improvement in two different ways: firstly, via ensuring better homogenization of cementitious pastes through the presence of citrus pectin (confirmed in SEM studies), and secondly, due to the formation of a double hybrid structure originating from (1) electrostatic interaction between polycationic chitosan and polyanionic pectin as well as (2) the hybrid nature of hydroxyapatite-chitosan granules. The existence of a double hybrid system resulted in better adhesion of the cement phase to the surface of the granules and thus the intimate contact of the components. The mechanical strength of developed materials allows their implantation in non-load or low-load bearing places (compressive strength of cancellous bone is approximately 4–12 MPa) [51]. Summarizing, the improved mechanical properties of the obtained multicomponent scaffolds could be attributed to the higher crosslinking degree promoted by multifaceted interactions between components.

### 2.6. Chemical Stability and Bioactivity In Vitro

The chemical stability of implantable biomaterials determines their potential clinical applications in the future. To evaluate the chemical stability and bioactivity in vitro, the biomicroconcretes were incubated in SBF at a temperature of 37 °C. The pH changes of SBF during the sample’s immersion are demonstrated in Figure 5. Despite the presence of polymeric additive, i.e., citrus pectin, as well as the hybrid HA/CTS granules, the pH values during the sample incubation ranged from 7.34 to 7.43 and remained close to the physiological pH [52].

After 7 days of incubation in SBF at 37 °C, all biomicroconcretes were fully covered by plate-like apatitic structures (Figure 6). The presence of apatitic forms confirmed the in vitro bioactivity of the developed materials according to the Kokubo and Takadama method [53]. The microstructure observations after the samples’ incubation revealed that the developed materials had bioactive potential. It can be stated that the application of polymeric additives, such as chitosan in the hybrid granules and pectin in the liquid phase, allowed obtaining materials with favorable physicochemical properties with bioactive potential.

The ionic conductivity of material MC1 was in the range of ~62–74 µS/cm. It was observed that the ionic conductivity of water around the incubated MC2–MC4 samples increased from ~65–81 µS/cm on the first day of incubation to ~80–91 µS/cm on day 28 (Figure 7).

The higher ionic conductivity of biomicroconcretes MC2, MC3, and MC4 compared to MC1 may have been related to the degradation of the citrus pectin. Pectin in aqueous solutions degrades to non-toxic products (i.e., d-galacturonic acid, l-arabinose, d-apiose, d-galactose, l-fructose) that do not have negative effects on cells when the material is implanted into the body [54,55]. Despite slight differences in the ionic conductivity of the distilled water surrounding them, the incubated samples of the biomicroconcretes all remained stable throughout the incubation time. Similar results regarding changes in the ionic conductivity of pectin-containing materials were previously reported elsewhere [56].

## 3. Materials and Methods

### 3.1. Materials

As a solid phase of the studied materials, α-tricalcium phosphate (α-TCP) and hybrid hydroxyapatite/chitosan (HA/CTS) granules were used. The highly reactive α-TCP powder was obtained via a wet chemical method described previously [21,57]. As the reagents, Ca(OH)_2_ (≥99.5%, POCH, Gliwice, Poland) and H_3_PO_4_ (85.0%, POCH, Gliwice, Poland) were applied. HA/CTS hybrid materials in the form of granules (300–400 µm) were prepared by the modified wet chemical method described by Zima [58]. Briefly, phosphoric acid was introduced directly to the chitosan solutions in acetic acid, and the obtained mixtures were added dropwise to the Ca(OH)_2_ suspension. The suspension was aged for 24 h and then decanted. The precipitate was washed with distilled water, centrifuged, and frozen for 48 h. After defrosting, the obtained filter cakes were sieved and dried to obtain hybrid granules. The following substrates were used: Ca(OH)_2_ (≥99.5 wt%, POCH, Gliwice, Poland), H_3_PO_4_ (85.0 wt%, POCH, Gliwice, Poland), and medium-molecular-weight chitosan (~100,000 kDa, DD ≥ 75.0 wt%, Sigma-Aldrich, St. Louis, MO, USA). As the liquid phases, various mixtures of 2 wt% disodium phosphate (99.9 wt%, Chempur, Piekary Śląskie, Poland) and 5 wt% low esterified amidated citrus pectin gels (Herbstreith & Fox, Werder/Havel, Germany) were applied. In order to optimize the setting time of the developed materials, Na_2_HPO_4_ was used in the liquid phase as an accelerator of the hydrolysis process, as recommended in other studies [59].

### 3.2. Preparation of Biomicroconcretes

In this study, five types of biomicroconcretes containing α-TCP powder and hybrid HA/CTS granules as a solid phase and different liquid phases were obtained (Table 3). Samples were prepared by mixing powder phase with the appropriate liquid phase. The liquid-to-powder (L/P) ratio was selected based on preliminary studies, and it was 0.5 g/g. Material MC1 containing solely 2 wt% disodium phosphate solution as a liquid phase was used as a control material. Subsequent materials in addition to disodium phosphate solution contained also a citrus pectin gel.

### 3.3. Methods

#### 3.3.1. Setting Times

The setting times of the obtained materials were measured. The initial (t_i_) and final (t_f_) setting times of the obtained biomaterials were determined using a Gillmore apparatus (Humboldt, Norridge, IL, USA) according to the ASTM-C200-08 standard [60]. All experiments were carried out at 22.0 ± 1.0 °C. Test samples were prepared in 10 mm × 7 mm × 3 mm cuboid form. All measurements were performed in triplicate. The results are presented as the mean ± standard deviation (SD).

#### 3.3.2. Injectability

The injectability of the obtained materials was assessed by injecting cement paste through the 2 mm nozzle of a 20 mL plastic syringe (B. Brown, Melsungen, Germany) directly into the cylinder with simulated body fluid (SBF) solution, preheated to 37 °C. The force applied to the syringe plunger during the injection was determined by a universal testing machine (Instron 3345, Instron, Norwood, MA, USA). The crosshead displacement rate was equal to 1.0 mm min^−1^. Measurements were performed in triplicate for each material. The value for the injectability of the material was considered to be the constant value of the pressure force during the extrusion of homogenous paste. The results are presented as the mean ± standard deviation (SD). Figure 8 shows the testing equipment for the injectability tests.

#### 3.3.3. Phase Composition

The crystalline phases of obtained biomicroconcretes were analyzed using X-ray diffraction (XRD) with Cu Kα radiation (1.54 Å) at 30 kV and 10 mA (D2 Phaser, Bruker, Billerica, MA, USA) within the 2θ range from 10 to 60° at 0.04 intervals (scanning speed of 2.5°/min). The analyses were performed after 7 days of setting and hardening in air, as well as after 7 days of incubation in simulated body fluid (SBF). The crystalline phases were identified by comparing the experimental diffractograms to the patterns from International Centre for Diffraction Data α-TCP (00-009-0348) and hydroxyapatite (HA; 01-076-0694). A phase quantification based on Rietveld refinement was performed using TOPAS software (Version 4.2.0.1, 2011, Bruker, Billerica, MA, USA). All measurements were performed in triplicate. The results are presented as the mean ± standard deviation (SD).

#### 3.3.4. Structural Analysis

Structural studies of the obtained biomaterials were performed using Fourier transform infrared (FTIR) spectroscopy, within the scanning range of 400–4000 cm^−1^ and resolution of 4 cm^−1^ using a BioRad FTS 6000 spectrometer (Vertex 70&70v, Bruker, Billerica, MA, USA). Positions of the FTIR bands were measured in accordance with the center of weight. The baseline correction, normalization, and spectral analyses were performed using Spectragryph software (v1.2.15, Friedrich Menges, Oberstdorf, Germany). The measurements were carried out after 7 days of setting and hardening in air, as well as after 7 days of incubation in simulated body fluid (SBF).

#### 3.3.5. Compressive Strength

For the mechanical tests, cylindrical samples 12 mm in height and 6 mm in diameter were prepared in Teflon molds. Samples were removed from the molds between their initial and final setting times and left for 7 days in air. After 7 days, the compressive strength of the samples was examined using a universal testing machine (Instron 3345, Instron, Norwood, MA, USA). Biomicroconcrete samples were subjected to uniaxial compression with a crosshead speed of 1.0 mm min^−1^. The results of the compressive strength tests are presented as the mean value of 15 measurements ± standard deviation (SD).

#### 3.3.6. Microstructure

Observations of the microstructure of the obtained biomaterials were performed using a scanning electron microscope (SEM, PhenomPure, Thermo Fisher Scientific, Waltham, MA, USA) in backscattered electron (BSE) mode at an acceleration voltage of 10 kV. Before the study, samples were coated with a thin gold layer to avoid overcharging (Manual Sputter Coater 108, Agar Scientific, Stansted, UK).

#### 3.3.7. Chemical Stability and Bioactivity In Vitro

To determine the in vitro chemical stability of cementitious materials, cylindrical samples (3 mm in height and 6 mm in diameter) were placed in plastic containers with 20 mL of SBF or distilled water and stored at 37 °C for 4 weeks. The ionic conductivity and pH of the solutions around the incubated samples were measured using a SevenCompact Duo pH/conductometer (Mettler Toledo, Columbus, OH, USA). In vitro bioactivity of the obtained biomicroconcretes was assessed via SEM observations of the apatite layers on the materials’ surfaces after the 7 day sample incubation in SBF.

#### 3.3.8. Statistics

The statistical analysis of obtained results was performed using a one-way analysis of variance (ANOVA) with a post hoc Tukey honestly significant difference (HSD) test for comparing multiple treatments (*—statistically significant difference between the results, *p* > 0.05). All analyses were performed with OriginPro software (version 2021, OriginLab Corporation, Northampton, MA, USA).

## 4. Conclusions

In this study, materials based on highly reactive α-TCP powder and hybrid hydroxyapatite-chitosan granules with the addition of citrus pectin were developed and examined. The use of citrus pectin introduced with the liquid phase allowed us to to obtain easily moldable, fully injectable calcium phosphate-based biomicroconcretes characterized by good cohesion and setting times in an acceptable range. Usually, biomicroconcretes containing granules or microspheres are not injectable. Through the careful selection of the liquid phase, we were able to obtain a paste with favorable rheological properties that did not clog the syringe during the injection. The presence of citrus pectin in the liquid phase significantly improved both the injectability and mechanical strength of the materials (from 5.75 MPa to 13.23 MPa). The unique properties of biomicroconcretes containing citrus pectin resulted from both the the dual setting system and the presence of the double hybrid system. The dual setting system originated from α-TCP hydrolysis and citrus pectin crosslinking in the presence of Ca^2+^ ions allowed us to obtain materials characterized by excellent cohesion and chemical stability, whereas the double hybrid system was due to the presence of hybrid granules and interactions between polycationic chitosan in hybrid granules and polyanionic citrus pectin. All developed biomicroconcretes revealed in vitro bioactivity, which makes them good candidates for further biological studies. The MC3 biomicroconcrete is considered to be the most promising among the studied materials. By applying easy-to-functionalize polymers such as chitosan and citrus pectin, further modifications of the proposed materials, for example, attaching other substances to the functionalized polymer groups, will be possible. This research confirmed the beneficial properties of the obtained biomicroconcretes and paves the way for further in vitro and in vivo studies.

## Figures and Tables

**Figure 1 ijms-24-13455-f001:**
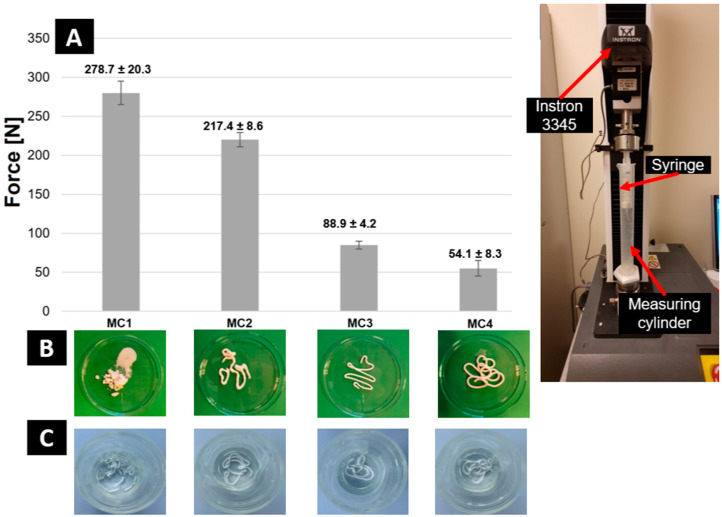
Force required for paste extrusion (**A**), pastes immediately after injection into SBF (**B**), and after 7 days of incubation in SBF (**C**).

**Figure 2 ijms-24-13455-f002:**
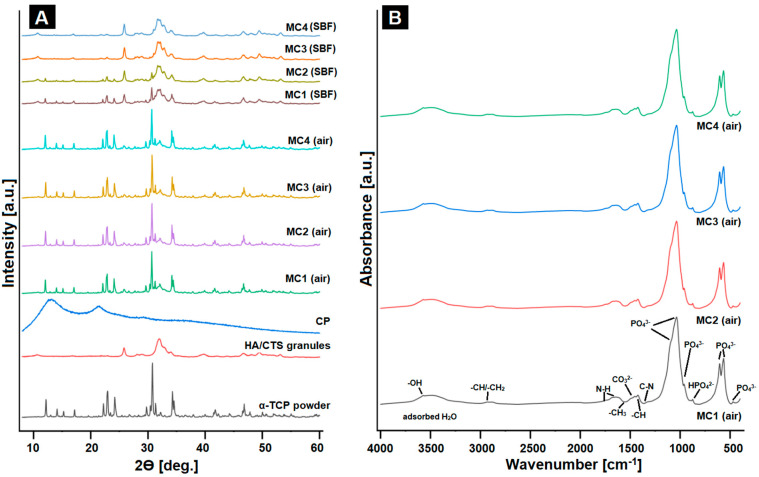
XRD patterns of studied materials (**A**) and FTIR spectra (**B**) of obtained biomaterials after 7 days of setting and hardening in air or in SBF.

**Figure 3 ijms-24-13455-f003:**
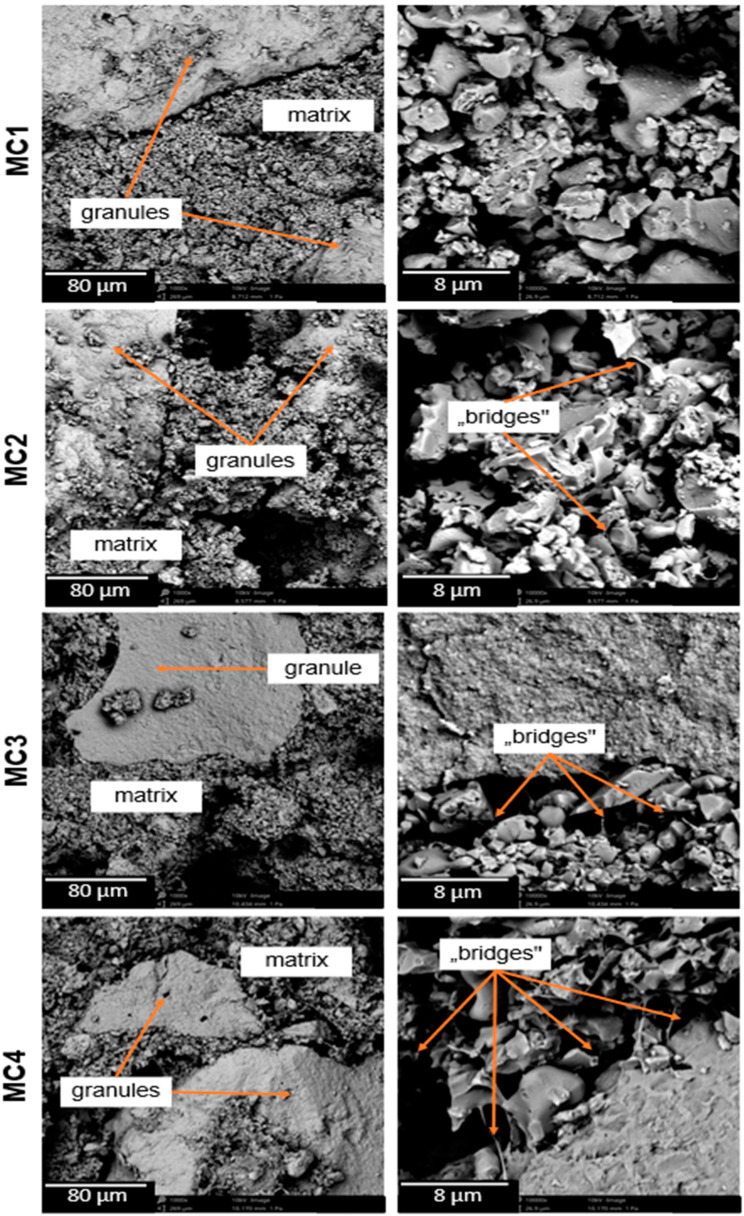
SEM images of the materials’ cross-sections after 7 day in air.

**Figure 4 ijms-24-13455-f004:**
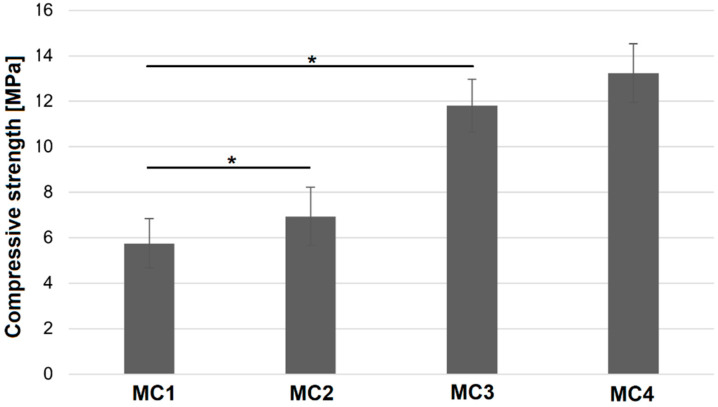
Compressive strength of biomicroconcretes after 7 days of drying in air (* statistically significant difference, *p* < 0.05).

**Figure 5 ijms-24-13455-f005:**
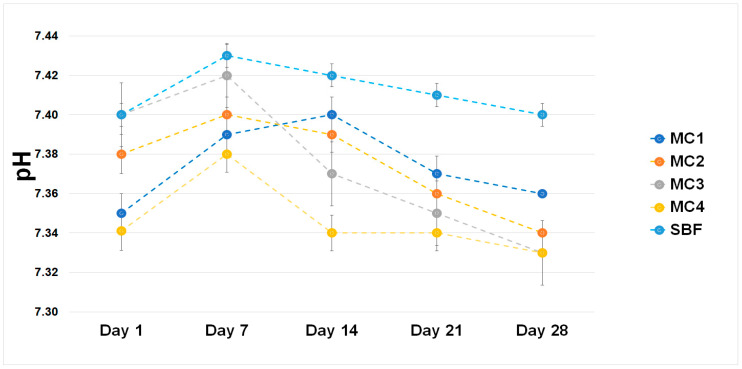
pH vs. sample incubation time in SBF.

**Figure 6 ijms-24-13455-f006:**
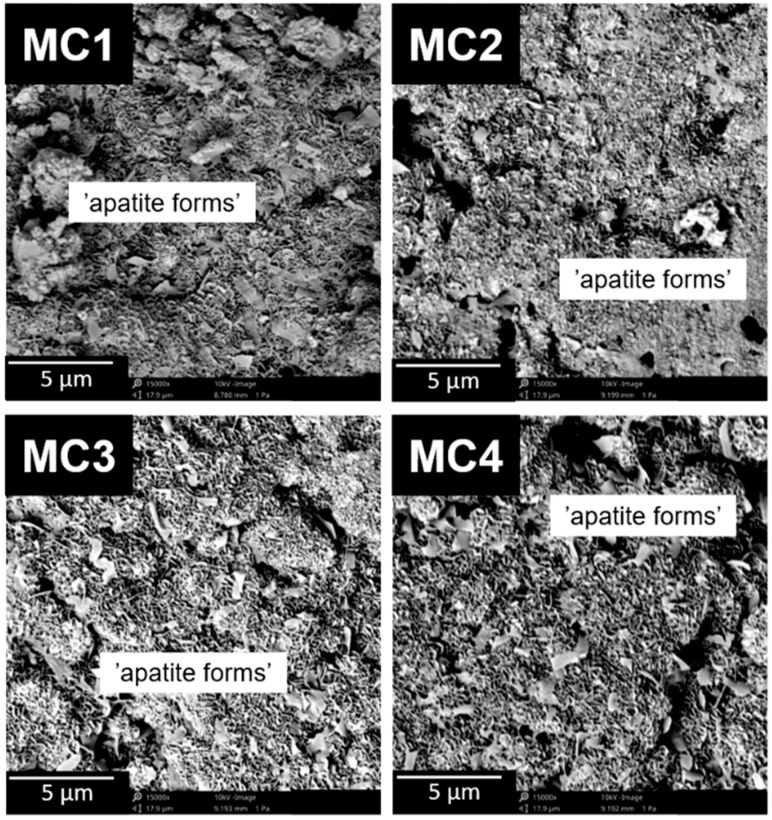
SEM microstructure of material surfaces after 7 days of incubation in SBF.

**Figure 7 ijms-24-13455-f007:**
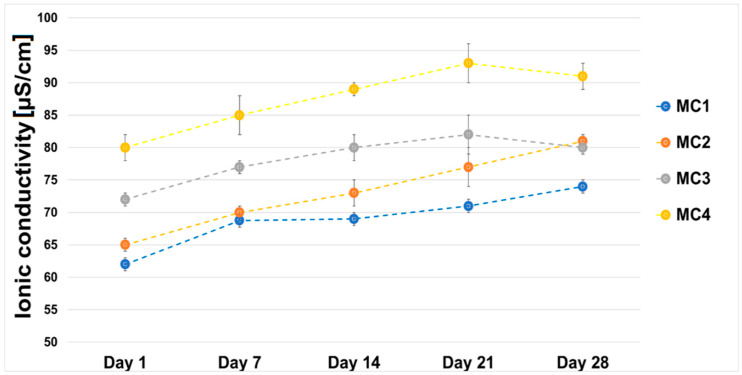
Ionic conductivity vs. sample incubation time in distilled water.

**Figure 8 ijms-24-13455-f008:**
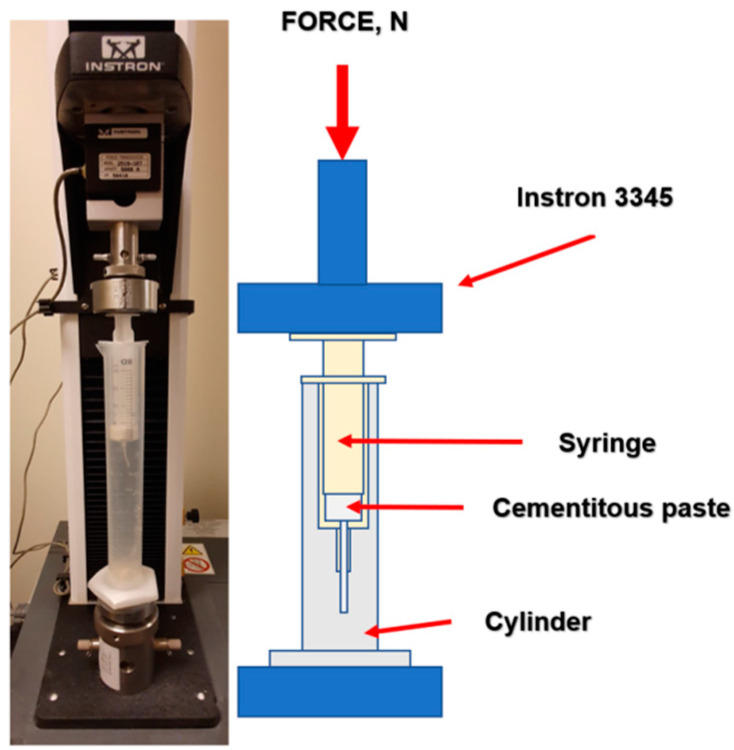
Injectability testing equipment.

**Table 1 ijms-24-13455-t001:** Setting time of the biomicroconcretes.

Material	Initial Setting Time (t_i_) [min]	Final Setting Time (t_f_) [min]
MC1	4.5 ± 1.0	7.5 ± 0.5
MC2	9.0 ± 0.5	16.5 ± 1.0
MC3	11.5 ± 0.5	21.0 ± 1.0
MC4	17.5 ± 1.0	32.0 ± 1.5
MC5	30.5 ± 0.5	55.5 ± 1.0

**Table 2 ijms-24-13455-t002:** Phase compositions of the obtained materials after 7 days of setting and hardening in air or SBF.

Material	7 Days in Air		7 Days in SBF	
	α-TCP, wt%	Hydroxyapatite, wt%	α-TCP, wt%	Hydroxyapatite, wt%
MC1	54 ± 1	46 ± 1	2 ± 1	98 ± 1
MC2	56 ± 1	44 ± 1	2 ± 1	98 ± 1
MC3	59 ± 1	41 ± 1	3 ± 1	97 ± 1
MC4	61 ± 1	39 ± 1	3 ± 1	97 ± 1

**Table 3 ijms-24-13455-t003:** The initial composition of developed materials.

Material	Solid Phase (P)	Liquid Phase (L)
MC1 (Control)	α-TCP: HA/CTS granules 3:2	2.0 wt% Na_2_HPO_4_ solution
MC2		1.5 wt% Na_2_HPO_4_ solution in 1.25 wt% citrus pectin gel
MC3		1.0 wt% Na_2_HPO_4_ solution in 2.5 wt% citrus pectin gel
MC4		0.5 wt% Na_2_HPO_4_ solution in 3.75 wt% citrus pectin gel
MC5		5.0 wt% citrus pectin gel

## Data Availability

Not applicable.

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
