# Peer review of "Novel Double Hybrid-Type Bone Cements Based on Calcium Phosphates, Chitosan and Citrus Pectin"

_ijms, 2023, doi:10.3390/ijms241713455_

Round 1

Reviewer 1 Report

The study of Pańtak et al. present an interesting use of nanohybrids in orthopaedic applications. Characterization regarding mechanical and chemical properties is well explained. Minor reviews were detected mainly, but we encourage authors to answer the questions asked for bioactivity and rheological topics. Language is very well writing (small details though), but it was a good work.

Abstract

Line 19: In vitro, it should be in italics.

It is suggested to mention results because there is nothing.

Line 26: Mention what TPC is because it is mentioned for the first time.

Line 29: Modify the reference to [2,3]. Review throughout the document.

Line 57: The word “pectin” is in a different type of letter.

Line 68: The word “stable” is repeated.

Line 75: What do you mean by “mers”?

Line 109: If, for all combinations, the solid phase was of 3:2 and L/P of 0.5, I suggest that this only remains in the text and not in Table 1.

Line 133: The figure should be centered.

Line 156: “of” after mechanical test is missing.

Line 200: In Table 2, “i” and “f” for time should be subscript. Also, include the statistical analysis to determine if significant differences were found.

Line 201: Space in 2.0wt% is missing.

Line 202: …an equal to 4.5 min and 7.5 min.

Line 206: It is recommended to use an equation editor for formulas.

Line 239: Center figure. Also, both graphs and photos could be bigger.

Line 263: Figure size could be improved.

Line 275: Sentence grammar should be checked.

Line 335: Center figure. I suggest highlighting axes.

Line 339: Space in 2.0wt% is missing.

Line 363: Center figure.

Line 374: Center figure. I wonder if the resolution could improve.

Line 376: Space is missing after MC1.

Line 380: Center figure.

Conclusion:

Authors mention that materials exhibit in vitro bioactivity, but an essay to prove this was not conducted. Also, favorable rheological properties are mentioned, but the viscosity or shear rate determination still needs to be explored.

Language is easily understandable and well written.

Author Response

Dear Reviewer,

Our response to the reviews is in the attached text document.

Best regards,

Authors

Reviewer 2 Report

The authors have made a commendable effort in investigating the influence of liquid phase composition on the physicochemical properties of double hybrid-type bone substitutes and developing double-hybrid materials with a dual setting system. While the manuscript is well-written, there are some concerns that need to be addressed before considering it for publication.

1.In the section titled "2.3.2. Injectability," it would be helpful to clarify whether phase separation was observed during extrusion from the syringe in different groups. Please provide additional details regarding any observed phase separation.

2.Regarding Figure 2, showcasing the pastes immediately after injection into simulated body fluid (SBF). However, it would greatly enhance the study if the authors could include images of the pastes after a 7-day incubation in SBF to assess the effect of SBF on the biomicroconcrete shape.

3.As the developed materials are categorized as a dual-setting system, it is essential for the authors to compare this material with existing systems to determine its advantages and unique features.

4.Please provide more comprehensive information on the composition of the biomicroconcretes based on X-ray diffraction (XRD) analysis. Specifically, elaborate on the specific components and their respective proportions in the biomicroconcretes.

5.The study encompasses the evaluation of compressive strength, chemical stability, and bioactivity under various conditions. For instance, ionic conductivity was measured during incubation in distilled water, while the structure was assessed during incubation in SBF. It would be valuable to provide an explanation for the use of different conditions and how they relate to the properties being measured.

6.In Figure 4, displaying SEM images of the materials' cross-section after a 7-day incubation in air, it would be beneficial to include control groups.

 Minor editing of English language required

Author Response

(The authors gave the same response as above.)

Reviewer 3 Report

The work entitled “Novel double hybrid-type bone cements based on calcium phosphates, chitosan and citrus pectin” is well structured and it contains interesting results on novel bone cements.

Anyway, some changes are needed to improve the quality of the work.

Firstly, the introduction section should be improved by including recent findings in the proposed filed, also better highlighting the most important findings in comparison with the recently published paper proposed by the same research team (Pantak et al, Materials 2021).

The improvements related to the use of the proposed novel bone substitute should be better highlighted in the abstract and Introduction section.

The presence of defects observed via SEM (figure 4) should be responsible for the failure or for bacterial infiltration in the cements sample. To this aim, some points related to the possibility to include antibacterial active agents to overcome issues related to mechanical properties and bacterial resistance to antibiotics should be underlined (see for example Russo et al., Polymers 2020; De Santis et al, Materials 2021).

To this aim, did the authors perform microbiological tests against bacterial strains commonly found in the oral mouth?  

Did the authors also evaluate shrinkage of the samples occurring after the setting time?

Regarding “Methods” section, the authors should make clearer the sentences regarding the adopted method for biomicroconcrete preparation. The authors could also include information regarding HA/CTS granule preparation as well as details regarding their morphology and bioactive features.

Regarding the injectability tests, the authors could include a table containing details on applied load to inject proposed systems, or alternatively, they could provide a correlation point (e.g. highlighting changes and trends related to the composition) in the discussion of the injectability results.

SEM images should be provided in a very better quality. Additionally, cross-section of proposed samples after compression tests should be also analysed via SEM, thus highlighting the eventual crack propagation onset.

Author Response

(The authors gave the same response as above.)

Round 2

Reviewer 2 Report

Thank you, authors, for addressing the issues raised.

Minor editing of English language required